# Electrochemical Sensing of Favipiravir with an Innovative Water-Dispersible Molecularly Imprinted Polymer Based on the Bimetallic Metal-Organic Framework: Comparison of Morphological Effects

**DOI:** 10.3390/bios12090769

**Published:** 2022-09-19

**Authors:** Nevin Erk, Mohammad Mehmandoust, Mustafa Soylak

**Affiliations:** 1Department of Analytical Chemistry, Faculty of Pharmacy, Ankara University, Ankara 06560, Turkey; 2Department of Chemistry, Faculty of Sciences, Erciyes University, Kayseri 38039, Turkey; 3Technology Research & Application Center (TAUM), Erciyes University, Kayseri 38039, Turkey; 4Turkish Academy of Sciences (TUBA), Ankara 06670, Turkey

**Keywords:** COVID-19, favipiravir sensing, molecularly imprinted polymer, Co/Ni metal–organic framework

## Abstract

Molecularly imprinted polymers (MIPs) are widely used as modifiers in electrochemical sensors due to their high sensitivity and promise of inexpensive mass manufacturing. Here, we propose and demonstrate a novel MIP-sensor that can measure the electrochemical activity of favipiravir (FAV) as an antiviral drug, thereby enabling quantification of the concentration of FAV in biological and river water samples and in real-time. MOF nanoparticles’ application with various shapes to determine FAV at nanomolar concentrations was described. Two different MOF nanoparticle shapes (dodecahedron and sheets) were systematically compared to evaluate the electrochemical performance of FAV. After carefully examining two different morphologies of MIP-Co-Ni@MOF, the nanosheet form showed a higher performance and efficiency than the nanododecahedron. When MIP-Co/Ni@MOF-based and NIP-Co/Ni@MOF electrodes (nanosheets) were used instead, the minimum target concentrations detected were 7.5 × 10^−11^ (MIP-Co-Ni@MOF) and 8.17 × 10^−9^ M (NIP-Co-Ni@MOF), respectively. This is a significant improvement (>10^2^), which is assigned to the large active surface area and high fraction of surface atoms, increasing the amount of greater analyte adsorption during binding. Therefore, water-dispersible MIP-Co-Ni@MOF nanosheets were successfully applied for trace-level determination of FAV in biological and water samples. Our findings seem to provide useful guidance in the molecularly imprinted polymer design of MOF-based materials to help establish quantitative rules in designing MOF-based sensors for point of care (POC) systems.

## 1. Introduction

The coronavirus (COVID-19) has been a strange shock in the twentieth century. Most people who became sick with COVID-19 experienced slight to typical symptoms and recovered without special treatment. Now, this virus has become a pandemic all over the world. Unfortunately, no specific therapeutic agent has been approved for the treatment of COVID-19 until now. Therefore, the discovery of a new and particular antiviral agent against SARS-CoV-2 would include a long and laborious timeline [1]. By default, several drugs have already been suggested for use against other viral infections (antimalarial drugs), which can positively affect COVID-19 treatment, such as chloroquine and hydroxyl-chloroquine [2,3]. Favipiravir (FAV, brand name: Favimol^®^) was introduced as an influential and successful candidate drug for COVID-19 treatment after the coronavirus pandemic in 2019. FAV undergoes renal excretion, eliminated in the urine mostly as a hydroxide, resulting in the plasma levels of this drug being challenging to control due to its once-daily dosing regimen [4]. Consequently, the ability to accurately detect FAV in the environment and biological samples is increasingly important.

Several analytical methods for FAV determination are commercially available, including high-performance liquid chromatography (HPLC) [5,6], spectrometric [7], and spectrofluorimetric methods [8]. Despite the significant advances that have been made by the above traditional approaches, a portable and reliable method for selective determination in the laboratory and environment still faces substantial challenges such as multiple interferences and personal care products, high cost, and the need for complicated and expensive instruments. In addition to the aforementioned methods, electrochemical techniques are applied to analyze electroactive analytes [9]. Indeed, because of their sensitivity, time savings, simple operation, and portability, electrochemical methods have been extensively studied for portable applications. Recently, Mohamed et al. [10] proposed an electrochemical sensor based on MnO_2_-rGO to determine FAV in dosage form and plasma samples. This developed method showed a wide linear concentration range of 0.01 up to 55.0 µM and a low limit of detection (LOD) and quantification limit of 0.11 and 0.33 µM, respectively. However, despite the high selectivity, the combination of manganese dioxide (MnO_2_) and reduced graphene oxide (rGO) did not indicate a significant limit of detection compared to our work. Allahverdiyeva et al. [11] provided an electrochemical method for favipiravir determination using boron-doped diamond (BDD) electrodes in the presence of cetrimonium bromide (CTAB). The electrode demonstrated two linear segments of 0.064–0.64 and 0.64–130.0 µM and an LOD of 0.018 µM. However, the BDD electrodes have some disadvantages, such as possible surface passivation [12], that may reduce the electrochemical activity of the developed electrode. Additionally, Wang et al. [13] proposed an electrochemical sensor based on molecularly imprinted polymer to detect FAV in biological samples using a core-shell nanocomposite of flower-like molybdenum disulfide nanospheres. This advanced electrode showed a wide linear response range of 0.01  to  100 nM and a low LOD of 0.002 nM. However, this approach to the preparation of the MIP sensor suffers from the use of dangerous solvents such as dimethylformamide (DMF), which may cause health hazards. To tackle these issues, green synthesis methods have been utilized to reduce the possible risks. The green approach can be characterized by it ease regarding nanoparticle production, cost-effectiveness, and eco-friendliness [14]. However, to our knowledge, no research has been performed to determine FAV based on water-dispersible molecularly imprinted polymer using the green approach.

The advantages of our novel approach encouraged our research group to construct voltametric-based procedures for assessing FAV in real samples. Moreover, a fast and sensitive method for trace FAV detection is still desirable, and despite these and other efforts in the literature [10,11,15,16], efficient, alternative methods are still needed to meet the requirements of a wide range of sensing applications.

The molecular imprinting approach is a strategy for designing and building sensors that have particular recognition for target molecules. In comparison to other functionalized materials, MIPs have distinguishing characteristics such as good stability, high affinity, ease of production, and cheap cost, and have therefore proven to be one of the most competitive tools in the field of biomolecule recognition [17]. MIPs in electrochemical sensors can either collect template molecules on the electrode’s surface, making the sensor more sensitive, or separate the template molecules from other analytes, improving the sensor’s selectivity. Because of these characteristics, MIPs are appropriate recognition components for sensors. However, MIPs are typically synthesized in organic solvents, whereas most of the practical applications of MIPs are assigned for the aqueous environment, including environmental monitoring, food safety control, and sensing applications. It has been demonstrated that the specific template binding of MIPs synthesized in organic solvents is significantly decreased in aqueous environments, which remarkably limits their applications in the biotechnology field [18,19]. To address this problem, enhancing the exchange surface by reducing the particle size can induce a better solubility efficiency in MIPs and use chitosan in acidic environments and conditions [20,21]. The second novelty of the present study is also the use of chitosan to synthesize water-soluble MIPs for the first time. 

Moreover, there are still many obstacles to the molecularly imprinted approaches, including the difficulty in electron transfer from the bulk solution to the imprinted sites during the recognizing process. As a result, molecularly imprinted polymers often indicate poor imprinted efficiency, low selectivity, and slow rebinding speed [22]. In this context, to overcome these obstacles, metal–organic frameworks have been expanded for the selective recognition of drugs. They have been demonstrated to improve the imprinted efficiency as modifiers in various biochemical and chemical sensors [23,24] because of their high catalytic and electrical properties, high chemical stability, sufficient sensitivity, and good active surface area. The prevailing involvement of MOF composite in a diversity of sensing technologies highlights the necessity for strategies that promote rapid determination and high selectivity in the presence of several interfering agents. MOF materials may give a short route between each component, improving electron transfer and avoiding the aggregation of individual components, providing more available active sites [25,26]. However, despite significant advances in the design and use of MOF-based materials, infrequent reports of modifiers remain. Interestingly, the integration of secondary metal nodes onto the MOF surface, resulting in bimetallic MOF, has demonstrated a high degree of porosity, stronger thermal stability, and improved intrinsic characteristics, and improved electronic conductivity and active site availability over monometallic MOF [27] because bimetallic bonds induce changes in the band structure of metals, therefore enhancing the stability and high electrocatalytic response towards FAV because of the synergistic effect of both metal ions [28,29]. 

To date, thanks to the recent developments in nanochemistry and nanomaterial engineering science, which allow multifunctional nanomaterials to be utilized in various application areas, including environmental [30], food [31], energy [32,33], biomedical [34], etc., combinations of nanomaterials and nanocomposites with MIPs are very popular and have already been introduced to solve poor binding and template leakage problems. Therefore, several aspects can be modified during a sensor design to improve the selectivity and sensitivity, such as the size and shape. The ideal combination of these factors can determine the success of the determination of analytes because particle geometry is a crucial parameter for sensing applications. However, it has not yet been reported whether the shape of the nanoparticles has any effect on the binding properties of molecularly imprinted polymers.

With this proviso in mind, in this work, we first report on the green synthesis of fabrication of a water-soluble molecularly imprinted sensor for stable and selective generation, which can respond to the presence of FAV. This study is also the first study to observe the effect of morphology on the electrochemical activity of Co/Ni@MOF with an MIP-based structure. Thus, our approach could provide a novel way to develop a sensing application. We synthesized two types of Co/Ni@MOF (nanosheets and nandodecahedron), which were designated as MIP-Co/Ni@MOF. Afterward, the electrocatalytic capability of these two MIP-Co/Ni@MOF was investigated using FAV as a representative analyte. The results indicated that MIP-Co/Ni@MOF nanosheets exhibit better electrochemical behavior than MIP-Co/Ni@MOF nanododecahedron because of the large surface area and more sites to produce MIP sensors. Inspired by this, we chose MIP-Co/Ni@MOF nanosheets for application to FAV determination. The feasibility of MIP-Co/Ni@MOF nanosheets on the screen-printed electrode (SPE) was observed and showed an ability to perform measurements in real samples. Herein, the developed sensor was illustrated to be a powerful analytical tool for nano-molar (0.075 nM) sensing of FAV in real samples.

## 2. Experimental Section

The detailed materials, apparatus, and methods are summarized in the electronic Appendix A. The basic preparation procedure is schematically shown in Figure 1.

## 3. Results and Discussion

### 3.1. Materials Characterizations 

The morphology and structure of Co/Ni@MOF nanosheets and nanododecahedron were characterized by SEM. Figure 1A (nanododecahedron) and C (nanosheets) show that the as-prepared Co/Ni@MOF have a diameter of about 500 and 100 nm, respectively, and an extraordinarily smooth surface with good dispersion and that of the MIP-Co/Ni@MOF nanosheets (Figure 1C) was about 250 nm, which suggested that the MIP films were coated on the surface of Co/Ni@MOF nanosheets. The smaller size of the imprinted nanoparticles can accommodate more template molecules and thus enhance the resulting MIP sensor’s sensing performance [35], suggesting that Co/Ni@MOF nanosheets could be more suitable as sensor modifiers. As shown in Figure 1A, Co/Ni@MOF has a 3D dodecahedral structure while its MIP-Co/Ni@MOF has a wrapped irregular spherical morphology (Figure 1B), and as can be observed, the surface of the film looks smooth and undistinguished, indicating the successful synthesis of MIP-Co/Ni@MOF. As shown in Figure 1D, an intensive polymer layer with a porous and selective structure was provided on the Co/Ni@MOF nanosheets surface (Figure 1D), indicating successful interaction between Co/Ni@MOF and chitosan for the synthesis of MIPs. Therefore, the structure of the Co/Ni@MOF nanosheets and intermolecular interaction between FAV and the imprinted active sites were well-maintained, thus improving the high sensing performance toward FAV in the selectivity, sensitivity, and stability in the nanosheet structure, and therefore MIP-Co/Ni@MOF nanosheets were used for further experiments. These results demonstrated that all particles were successfully synthesized. The uniform distribution of cobalt, nickel, carbon, and oxygen on the surface of Co/Ni@MOF can be observed through the EDX image. The EDX spectrum (Appendix A) clearly revealed that only the elements Co, Ni, C, and O are detected in the MIP-Co/Ni@MOF, indicating no other impurity element was observed and the successful synthesis of Co/Ni@MOF. Moreover, the corresponding EDX elemental mappings (Appendix A) confirm the successful MIP-Co/Ni@MOF composite synthesis. The distributions of C, O, Ni, and Co match the structure of MIP-Co/Ni@MOF well. Furthermore, AFM images can be used to directly observe the topography, porosity, and surface roughness of the MIP-Co/Ni@MOF nanosheets. Figure 1E,F exhibit the AFM analysis of the MIP-nanocomposite surface before and after template removal. Obviously, the surface of the MIP-Co/Ni@MOF before washing shows many circular structures and the development of the MIP layer after template removal makes the surface of the composite rougher, confirming the exit of the chitosan and the presence of the FAV template sites, indicating that the AFM images strongly corroborate the successful synthesis of MIP-Co/Ni@MOF. This observation is also similar to what was observed in the SEM analysis. These aforementioned results of the surface and elemental characterizations unveil the successful formation of MIP-Co/Ni@MOF.

Further evidence of the interaction between materials was investigated by FT-IR spectroscopy, as shown in Figure 2A. The wide absorption peak at ∼3430 cm^−1^ is assigned to the stretching vibration of −OH, which affirms the presence of the intramolecular hydrogen bonds in Co*_x_*Ni*_y_*-MOF. The predominant peaks at ∼1345 and ∼1635 cm^−1^ are assigned to the asymmetric and symmetric stretching vibrations of −COOH, respectively. The absorption peaks at 1628 and 1172 cm^−1^ are related to the bending vibration of NH_2_ and C–N stretching vibration, respectively [36], indicating the presence of chitosan bonds. Furthermore, extra absorption signals under 1000 cm^−1^ (500 cm^−1^) are identified as Ni-OH and Co–OH stretching and bending vibration, indicating that the composite was successfully synthesized [37,38]. The XRD pattern of MIP-Co/Ni@MOF is displayed in Figure 2B (5.0° to 90.0°), and the composition and crystallographic structure of MIP-Co/Ni@MOF were observed. As shown in Figure 2B, the main diffraction peaks located at 11° and 17° are assigned to the (100) and (101) planes of Co/Ni@MOF, respectively, accompanied by a diffraction peak at 34 [39] with JCPDS card No. 15-0806 for Co and 65-2865 for Ni [40]. The other peaks are around the 21.4° and 28.2° planes of chitosan, indicating the presence of chitosan [41,42], approving the successful synthesis of the composite. These results are in agreement with the FT-IR analysis shown in Figure 2A.

### 3.2. Adsorption Test

To verify the active sites on the MIP-Co/Ni@MOF nanosheets, an adsorption study was conducted by suspending 20 mg of MIP-Co/Ni@MOF and NIP-Co/Ni@MOF into 10.0 mL of FAV solution at an initial concentration (40.0 μg mL^−1^). After magnetic stirring at 25 °C for 30 min, the residual concentrations of FAV were observed by UV-Vis (Figure 3). As a result, the current density of the MIP-Co/Ni@MOF was remarkably decreased compared to NIP-Co/Ni@MOF due to the noteworthy accessibility to the specific binding sites and, afterward, quick diffusion of target analytes into them. In addition, the adsorption equilibrium of analytes on MIP was faster than that on NIP, most likely due to the difference in their binding sites, while the respective NIP only has one type of nonspecific binding site that completely differs from MIP. The thin and uniform MIP-Co/Ni@MOF nanosheets offered great mass transfer, conquering some of the disadvantages of bulk MIPs and nonthin materials [43]. These results also reaffirm the successful synthesis of MIP-Co/Ni@MOF.

### 3.3. Electrochemical Behavior of Electrode

Recently, more and more researchers have paid attention to MIP materials in expectation of their synergetic magnification of optical, electrical, or catalytic properties. However, to the best of our knowledge, there is no research comparing MIP materials’ morphological effects on sensing applications. It is self-evident that this work is significant in its evaluation of the different morphological effects, providing new ideas for other research works.

Appendix A presents the DPV curves of the MIP-Co/Ni@MOF nanosheets and MIP-Co/Ni@MOF nanododecahedron recorded during the electrochemical oxidations of FAV. As shown in Appendix A, the MIP-Co/Ni@MOF nanosheets showed remarkable performances with faster electron tomography kinetics (indicated by the smaller E_p_^ox^ values). This result confirmed the excellent electrocatalytic properties and large active surface area of MIP-Co/Ni@MOF nanosheets compared to the MIP-Co/Ni@MOF nanodedecahedron. The MIP-Co/Ni@MOF nanosheets were chosen for further experiments.

Moreover, after choosing the Co/Ni@MOF nanosheets, typical CVs of BSPE, Co/Ni@MOF/SPE, NIP-Co/Ni@MOF/SPE, and MIP-Co/Ni@MOF/SPE (after template removal and rebinding) were examined in 5.0 mM [Fe(CN)_6_]^3^^−^^/4−^ as a redox probe containing 0.1 M KCl. As shown in Figure 4A, the redox peak of NIP-Co/Ni@MOF/SPE was weaker than that of MIP-Co/Ni@MOF/SPE, which was assigned to the lack of activity imprinting sites on the surface. However, Co/Ni@MOF/SPE enhanced the conductivity, and the current was more significant than that of the bare electrode. The many cavities for recognition sites on the imprinted film surface were matched with the spatial configuration of the imprinted molecules and fixed to the electrode surface, resulting in the formation of channels of the probe molecule [Fe(CN)_6_]^3^^−^^/4−^ through the imprinting holes. As a result, probe molecules diffused quickly on the electrode surface, facilitating an oxidation-reduction process, and enhancing the current response. Moreover, the voltammograms indicated a peak-to-peak separation (ΔEp) of 0.19 V at the bare electrode, and this value was lower at Co/Ni@MOF/SPE (0.16 V). The observed decrease in the ΔEp value at the Co/Ni@MOF/SPE electrode may be attributed to a greater charge-transfer process on the Co/Ni@MOF/SPE than on the bare electrode, which may be attributed to possible interaction between the binding sites. The rebinding of FAV to the imprinted sites reduced the number of cavities accessible for the redox probe to reach the electrode surface, causing the redox signal to diminish once more. Together with those obtained from the comparison, these results confirm that MIP-Co/Ni@MOF/SPE has the best electroactivity to determine FAV.

EIS is commonly used to investigate the electron transport kinetics at developed electrodes. EIS typically consists of a semicircular component at higher frequencies and a linear component at lower frequencies, corresponding to the restricted and diffusion processes, respectively. The charge transfer resistance (Rct) of the [Fe(CN)_6_]^3−/4−^ redox probe is shown by the semicircle diameter. Hence, EIS curves were performed to investigate the electron transfer occurring at the electrodes’ surface to affirm the MIP electrode’s resistance during binding, elution, and rebinding processes (conditions: open-circuit potential: 0.1 V, frequency: 0.1 to 100 kHz). The electrochemical impedance behavior of the sensor under different conditions is consistent with that of CV. Owing to the poor conductivity of the MIP membrane, MIP-Co/Ni@MOF/SPE showed a large resistance. After reabsorption of the FAV, the imprinted cavities were occupied again, indicating that the occupation of MIP cavities blocks the electron transfer by the FAV, providing specific information recognition toward FAV (Figure 4B). All the abovementioned results of the EIS, CV, AFM, SEM, XRD, and FT-IR data during the preparation steps were evidence of the successful manufacturing of MIP-Co/Ni@MOF/SPE regarding selective adsorption and FAV determination.

### 3.4. Impact of pH and Scan Rate

The CV responses of MIP-Co/Ni@MOF/SPE at different pH were investigated to reveal the effect of the pH valve on the peak current. As shown in Appendix A, the highest oxidation current of favipiravir at pH 4.0 was observed due to the tailor-made imprinted cavities and the stronger noncovalent binding at pH 4.0. Therefore, pH 4.0 was chosen as the best and optimal pH for further studies. At pH above 4, a significant decrease was observed at the modified electrode response, suggesting hydrolysis of FAV at higher pH, and FAV adsorption might be decreased on the electrode surface. Moreover, the linear regression plots were observed for the pH values vs. oxidation potential E_pa_ = −0.03 pH + 1.28 (*R*^2^ = 0.989). According to the Nernst equation (dE_p_/dpH = 0.059 m/αn), the m/n (proton number/electron number) ratio was obtained at 1:2. Hence, the possible electro-oxidation mechanism for FAV at MIP-Co/Ni@MOF/SPE is exhibited in Figure 2.

The effect of the scan rate on the peak current of FAV was also investigated to recognize the kinetics of the electrochemical process. Similarly, to visualize the changes, the CV curves at MIP-Co/Ni@MOF/SPE under various scan rates of 10.0 to 500.0 mV/s are displayed in Figure 5B. The peak potentials shifted to a more positive potential with the increase in the scan rate, verifying the irreversible character of the electrode reaction. A linear relationship was observed between the peak current (I_pa_) and the square root of the scan rate (*v*^1/2^): I = 0.325 *v*^1/2 Z^ + 0.21 (*R*^2^ = 0.995) (Appendix A inset), indicating that the process of the FAV at the developed electrode is diffusion-controlled. Also, the slope obtained from the graph of log I_pa_ as a function of log *v* is 0.498, which is very near to the generally expected slope value of 0.5, showing a purely diffusion controlled system (Appendix A). It is also observed that the oxidation peak potential (E_pa_) of FAV shifted towards the positive potential with the enhancement of the scan rate, suggesting an irreversible electrode reaction. Moreover, according to Laviron’s theory, the plot of the peak potential (E_pa_) against the natural logarithm of the scan rate (ln *ν*) shows a line (Appendix A), and using the slope of this line, the number of electron transfers was estimated to be approximately 2.0, indicating that the oxidation reaction of FAV at MIP-Co/Ni@MOF/SPE has two electrons transfer processes [15]. Therefore, the results obtained in the scan rate and pH scan, which affirm each other in the number of electrons transferred, reaffirm the mechanism of oxidation in Figure 2. The essential part of studying the mechanism of oxidation is the effect of this action on the effectiveness of FAV in the human body, and without knowing the underlying mechanisms, it is completely impossible to evaluate the exact function of the drug. The inactive drug (FAV) that enters the body after oxidation will be able to perform its reactions and show a practical effect (favipiravir-RTP). Therefore, it is crucial to know the oxidation mechanism of FAV.

### 3.5. Chronoamperometry Study

The value of the diffusion coefficient of FAV was estimated at the MIP-Co/Ni@MOF/SPE surface by the chronoamperometric method with an applied potential of 1.2 V (Appendix A). The Cottrell plots of 200 and 300 μM FAV are presented in Appendix A, and the diffusion coefficient of FAV was observed to be ∼1.28 × 10^−4^ cm^2^ s^−1^ by the Cottrell slopes of the Cottrell equation.

### 3.6. Analytical Application

The optimized conditions were used independently for the electroanalytical detection of FAV at both MIP-Co/Ni@MOF nanosheets and NIP-Co/Ni@MOF nanosheets using the DPV technique. As shown in Figure 5A,B, the FAV peak current increased gradually with the addition of FAV under optimal conditions. Moreover, as shown in the insets (Figure 5A,B), the relationship between the peak current and FAV concentration from 0.01 to 14.64 μM with a limit of detection of 8.17 nM for NIP-Co/Ni@MOF and from 0.1 to 151 nM with a limit of detection of 0.075 nM for MIP-Co/Ni@MOF was described. As shown, the sensitivity of MIP-Co/Ni@MOF/SPE compared to NIP-Co/Ni@MOF/SPE increased by approximately three orders of magnitude (sensitivity from 0.187 to 210.1 μM).

**Figure 5 biosensors-12-00769-f005:**
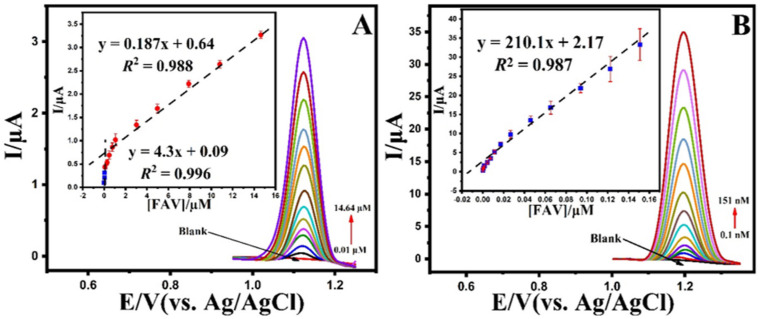
DPVs obtained at the (**A**) NIP-Co/Ni@MOF/SPE and (**B**) MIP-Co/Ni@MOF/SPE in 0.1 M B-R buffer (pH 4.0) in the various concentration (0.01 to 14.64 μM for NIP-Co/Ni@MOF/SPE and 0.1 to 151.0 nM for MIP-Co/Ni@MOF/SPE) of the FAV (insets). The calibration plot of current change vs. concentration of FAV. Condition; modulation amplitude: 0.05 V, modulation time: 0.01 s, interval time: 0.1 s.

The electrochemical response in the low concentration range was adequate for exhibiting the sensor’s high sensitivity towards favipiravir as a suitable approach in POC systems. Moreover, some recent reports on FAV electrochemical sensors are listed in Table 1 First, when compared to previous analytical methods, the modified electrode in this study shows good selectivity and sensitivity to detect FAV. Furthermore, when we look at the qualities of repeatability, reproducibility, and selectivity, the modified sensor in this work is more appropriate for FAV analysis. Hence, the results confirm that MIP-Co/Ni@MOF/SPE indicates a comparative detection limit and linear range for the detection of FAV.

### 3.7. Investigation of Selectivity

Selectivity is a crucial argument for estimating the performance of an electrochemical sensor because there is a considerable amount of co-existence and iso-structural species in clinical and biological specimens, which may cause interference with the target analyte. To verify whether the proposed sensor is specific towards FAV determination, the current performances of the coexistent interfering substances, including ascorbic acid, dopamine, uric acid, glucose, and l-cysteine, were investigated. The tolerance limit for FAV in the presence of 100-fold interferences was recognized as the maximum concentration with an RSD less than ± 7.0%. A slight change was also observed from some interferents (Appendix A) either due to hydrogen bonds’ weak interaction or coordination bonding with the hydroxyl group. Therefore, the results show that these substances did not interfere with favipiravir, confirming the proposed method’s selectivity.

### 3.8. Sensor Properties Investigation

The stability, reproducibility, and repeatability of MIP-Co/Ni@MOF/SPE were observed by the DPV method for 0.01 µM of FAV. The repeatability of a developed sensor of MIP-Co/Ni@MOF/SPE was performed using the same electrode 10 successive times under optimal conditions, and the RSD obtained was 1.4%. The reproducibility of the sensor was observed by the measurement of the current intensities when five different sensors were fabricated under the same condition. Consequently, the RSD value of the fabricated electrode was also calculated at 1.8%, indicating good sensor-to-sensor reproducibility. Moreover, MIP-Co/Ni@MOF/SPE stability was also observed using 0.01 µM of FAV under optimal conditions. The sensor was dipped in 0.1 M B-R buffer at 4 °C when not in use to ensure stability. No apparent decrease was observed over 5 days (RSD of 2.29%). After 5 days, it decreased by about 16.9% compared with the initial response. The decrease in the signal might cause degradation of FAV on MIP-Co/Ni@MOF/SPE. 

### 3.9. Analysis of Favipiravir in Real Samples

To assess the analytical performance of the proposed sensor, the MIP-Co/Ni@MOF/SPE was performed to determine FAV in river water, urine, and human plasma (Dyna-Tek Industries Inc., Lenexa, KS, USA). A standard voltametric strategy was performed using prepared solutions of river water, human plasma, and urine samples. As shown in Table 2, the recovery test was observed, indicating that this developed sensor is suitable for detecting FAV in real samples.

## 4. Conclusions

A novel sensing approach based on a molecularly imprinted metal–organic framework was successfully developed. It took advantage of the voltammetry approach, MOFs, and molecular imprinting method, which can give a fast signal response for identification of FAV. Additionally, the developed sensor demonstrated an outstanding wide linear range (0.0001–0.151 µM), high sensitivity (210.1 µA/µM), selectivity, repeatability, and reproducibility with a low detection limit (0.000075 µM). It is concluded that this electrode could be used as a promising approach to detect FAV in real samples with satisfactory recovery (95.6–109.9%). Moreover, this work is the first attempt to develop an electrochemical sensor based on water-dispersible MIP to determine favipiravir. Furthermore, the developed electrode offers a simple, inexpensive electrode modifier with excellent sensitivity in various real samples. Our results showed that the particle shape is a significant factor that needs exploration to design an effective sensor for sensing applications. Finally, MIP-Co/Ni@MOF/SPE is suggested as a powerful and fast analytical tool for monitoring FAV in real samples.

## Data Availability

Not applicable.

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
