# Peer review of "Electrochemical Sensing of Favipiravir with an Innovative Water-Dispersible Molecularly Imprinted Polymer Based on the Bimetallic Metal-Organic Framework: Comparison of Morphological Effects"

_biosensors, 2022, doi:10.3390/bios12090769_

Round 1

Reviewer 1 Report

In this manuscript, Erk et al. demonstrated that a MIP@MOF-based sensor can measure electrochemical activity of the antiviral drug favipiravir. Two morphologies of this composite sensor were tested and the sensor is show to be effective in aqueous environments. The electrochemical experiments detailed here are well thought out and do indicate significant promise for this type of sensor. After some minor revisions to this manuscript, specifically to strengthen the background and explicitly state more clearly the general explanations of concepts that the authors have inferred from the reader, I would recommend this manuscript to be published in Biosensors.

1. Line 45: The connection between the necessity to detect FAV in environment vs. the use of this drug is not clear. What is the motivation?

2. Line 58-59: What is the current limit of detection with standard techniques. How does that compare to current state of the art non-traditional techniques? The authors denote a relation here to their work but have not quantified to support these claims. 

3. Line 70-73: These advantages may need to be emphasized before this claim. What is the “green approach”? 

4. Line 73-74: The authors note “a fast and sensitive method for trace FAV detection is still desirable”. How fast and how sensitive does this detection need to be? What is the motivation behind these metrics? Are there regulations surrounding excessive amounts?

5. Line 121-124: Why was this particular MOF chosen?

6. Figure S2: It is unclear what the original sample being imaged here is. The original micrograph without color distribution should be provided for comparison.

7. Figure 3: A legend describing differences between colored spectra is missing.

8. Section 3.2: The UV-Vis data shows qualitative decrease of peaks at specific wavelengths. The authors have made a lot of inferences based on peak decreases that need to be more carefully connected. How are specific peaks related to concentrations of FAV? The relationship between concentration of FAV and current density of the MIP needs to be further explained. Moreover, this adsorption test has not been quantified. It may be possible to add a reference standard in solution at a known concentration in order to perform quantitative peak-peak analysis.

9. Line 225-226: The relationship between “electron tomography kinetics” and the DPV curves is unclear.

10. Line 350-352: The authors do not show this data comparing concentrations of FAV in relation to other interferences.

11. Line 367-368: The signal response is reported to have decreased nearly 17% compared with initial response after just 5 days. What part of the sensor is degrading during this time? What implications does this have on the overall effectiveness of the sensor?

Author Response

#Reviewer:1

Thank you for the comments on our manuscript entitled "Electrochemical sensing of Favipiravir with an innovative water-dispersible molecularly imprinted based on the bimetallic metal-organic framework: Comparison of morphological effects" We appreciate the suggested modifications and have revised the manuscript accordingly. Furthermore, the authors have modified the article in the electrochemical sections to show a better result. The revised sections are shown in yellow type. The detailed responses to the reviewers’ comments are presented as follows:

In this manuscript, Erk et al. demonstrated that a MIP@MOF-based sensor can measure electrochemical activity of the antiviral drug favipiravir. Two morphologies of this composite sensor were tested and the sensor is show to be effective in aqueous environments. The electrochemical experiments detailed here are well thought out and do indicate significant promise for this type of sensor. After some minor revisions to this manuscript, specifically to strengthen the background and explicitly state more clearly the general explanations of concepts that the authors have inferred from the reader, I would recommend this manuscript to be published in Biosensors.

  1. Line 45: The connection between the necessity to detect FAV in the environment vs. the use of this drug is not clear. What is the motivation?

Answer: Thank you so much for your great attention. I revised this sentence and the biological sample was also added in the revised version. As we mentioned in the prior sentence, because of the presence of FAV in human urine after using, this drug will release in the environment such as water; so water sample was chosen as real samples.

  1. Line 58-59: What is the current limit of detection with standard techniques. How does that compare to current state of the art non-traditional techniques? The authors denote a relation here to their work but have not quantified to support these claims.

Answer: Thank you so much for this point. You are completely right. The comparison between previously published articles and our work was added in the revised version.

  1. Line 70-73: These advantages may need to be emphasized before this claim. What is the “green approach”?

Answer: Thank you so much for your contribution. As the reviewer requested, the advantage of green methods was discussed in the revised version.

  1. Line 73-74: The authors note “a fast and sensitive method for trace FAV detection is still desirable”. How fast and how sensitive does this detection need to be? What is the motivation behind these metrics? Are there regulations surrounding excessive amounts?

Answer: Thank you so much for your great attention and valuable time. As we mentioned before, there are several methods to detect FAV in real samples; however, they are not fast enough, selective, sensitive, as well as environmentally friendly. Therefore, we mentioned that there is an urgent need to better detect FAV in real samples.

  1. Line 121-124: Why was this particular MOF chosen?

Answer: Thank you so much for this point out. As you know, the MOF materials have a wide surface area that makes them appropriate to use in sensing applications. However, these types of materials suffer from high conductivity, which is the main property that is needed for electrochemical sensors. Therefore, we chose this bimetallic MOF to enhance the conductivity of our developed electrode.

  1. Figure S2: It is unclear what the original sample being imaged here is. The original micrograph without color distribution should be provided for comparison.

Answer: Thank you so much for your attention. We wrote in the caption that is the image of MIP-Co/Ni-MOF and also we mentioned the elements of Ni and Co in the images. These images are completely original. I hope we have understood your meaning correctly.

  1. Figure 3: A legend describing differences between colored spectra is missing.

Answer: Thank you so much for your contribution. These two figures show the difference in absorption between MIP and NIP sensors that shows the adsorption efficiency of MIP is higher than NIP. We mentioned that A is for MIP and B is for NIP sensors that are difference completely.

  1. Section 3.2: The UV-Vis data shows qualitative decrease of peaks at specific wavelengths. The authors have made a lot of inferences based on peak decreases that need to be more carefully connected. How are specific peaks related to concentrations of FAV? The relationship between the concentration of FAV and current density of the MIP needs to be further explained. Moreover, this adsorption test has not been quantified. It may be possible to add a reference standard in solution at a known concentration in order to perform quantitative peak-peak analysis.

Answer: Thank you so much for this point out. You are entirely right. However, we used a UV study to approve the successful synthesis of the MIP sensor after elution. As you can be seen, the decreased intensity in MIP particles is higher than in NIP, indicating that more cavities have been created on the MIP materials and can be removed after elution. These properties cause to absorption of more FAV molecules from the solution which causes the higher density of synthesized MIP. As shown in this figure, after adding 20.0 mg of MIP to the FAV solution the peak decreased, in this section we just wanted to approve the successful synthesis procedure of MIP.

  1. Line 225-226: The relationship between “electron tomography kinetics” and the DPV curves is unclear.

Answer: Thank you so much for your contribution and valuable time. As you know, there is the main relation between the peak potential and kinetic of the developed sensor. As you can see, the peak position (Epa) of MIP-Co/Ni@MOF nanosheets appeared before the MIP-Co/Ni@MOF nanododecahedron, indicating the with faster electron tomography kinetics nanosheets compare to nanododecahedron nanoparticles.

  1. Line 350-352: The authors do not show this data comparing concentrations of FAV in relation to other interferences.

Answer: Thank you so much for your contribution and the authors think that your request will increase the quality of our work. As the dear reviewer requested, the DPV of FAV and interferences were added in supporting information (Fig. S6).

  1. Line 367-368: The signal response is reported to have decreased nearly 17% compared with initial response after just 5 days. What part of the sensor is degrading during this time? What implications does this have on the overall effectiveness of the sensor?

Answer: Thank you so much for your attention. You are completely right. The stability of our work is not sufficient because the MIP surface may fill and block in the presence of FAV and after a few days the current decreased remarkably.

Reviewer 2 Report

Journal: Biosensors

Article title: Electrochemical sensing of Favipiravir with an innovative water-dispersible molecularly imprinted based on the bimetallic metal-organic framework: Comparison of morphological effects

In the current manuscript, the authors discussed sensors about COVID-19. In the post-pandemic time, any research related to SARS-Cov-2 will greatly benefit society. The authors developed the sensors to detect Favipiravir drug which can be used as an antiviral drug related to COVID-19 treatment. The interesting part of the work is the authors used the bimetallic MOF for sensing applications. This is very important research in the area of sensors and its scope matches the MDPI biosensors journal. But the article needs some major revisions to get published in MDPI biosensors. 

I recommend its publication with major revision.

1.                   In the introduction authors need to discuss the advantages of the electrochemical sensors for sensing Favipiravir over other classical analytical methods.  

2.                   Literature review is poor, the authors need to discuss the importance of the synergy in the bimetallic MOF in electrochemistry I recommend authors make a separate paragraph on it here I am mentioning some of the articles the authors need to discuss them. It will be very much helpful for readers

a)       https://www.sciencedirect.com/science/article/abs/pii/S0010854522003770

b)      https://www.sciencedirect.com/science/article/abs/pii/S0013935121016212

c)       https://pubs.rsc.org/en/content/articlelanding/2022/an/d1an01978c/unauth

d)      https://www.sciencedirect.com/science/article/abs/pii/S0045653522010098

e)      https://www.sciencedirect.com/science/article/abs/pii/S0304389418301092

3.                   How the authors predicted the structure of MOF needs to be explained

4.                   Why the authors have not performed the XPS analysis to know the oxidation states of bimetals?

5.                   Why authors selected Co and Ni-based MOF

6.                   Is the oxidation potential of Co and Ni affecting the FAV oxidation reactions needs to be discussed

7.                   SEM results are good but further discussion is necessary related to the bimetallic presence

8.                   Respective Bode plot for Fig.4b need to be shown

9.                   What’s the use to detect FAV in river water samples needs to be explained

10.               What is the active surface area of MOF-based electrodes

11.               Why do the authors use CV over DPV?

12.               The conclusion needs to be elaborated.

13. Cross-check the references

I feel the theme of the article is interesting and attractive. 

Author Response

#Reviewer:2

Thank you for the comments on our manuscript entitled "Electrochemical sensing of Favipiravir with an innovative water-dispersible molecularly imprinted based on the bimetallic metal-organic framework: Comparison of morphological effects" We appreciate the suggested modifications and have revised the manuscript accordingly. Furthermore, the authors have modified the article in the electrochemical sections to show a better result. The revised sections are shown in yellow type. The detailed responses to the reviewers’ comments are presented as follows:

In the current manuscript, the authors discussed sensors for COVID-19. In the post-pandemic time, any research related to SARS-Cov-2 will greatly benefit society. The authors developed the sensors to detect Favipiravir drug which can be used as an antiviral drug related to COVID-19 treatment. The interesting part of the work is the authors used the bimetallic MOF for sensing applications. This is very important research in the area of sensors and its scope matches the MDPI biosensors journal. But the article needs some major revisions to get published in MDPI biosensors. 

I recommend its publication with major revision.

1.In the introduction authors need to discuss the advantages of the electrochemical sensors for sensing Favipiravir over other classical analytical methods. 

Answer: Thank you so much for your contribution. We discussed the advantage of the electrochemical sensor compared to traditional methods. The electrochemical sensors are more selective, sensitive, and low cost.

  1. Literature review is poor, the authors need to discuss the importance of the synergy in the bimetallic MOF in electrochemistry I recommend authors make a separate paragraph on it here I am mentioning some of the articles the authors need to discuss them. It will be very much helpful for readers
  2. a) https://www.sciencedirect.com/science/article/abs/pii/S0010854522003770
  3. b) https://www.sciencedirect.com/science/article/abs/pii/S0013935121016212
  4. c) https://pubs.rsc.org/en/content/articlelanding/2022/an/d1an01978c/unauth
  5. d) https://www.sciencedirect.com/science/article/abs/pii/S0045653522010098
  6. e) https://www.sciencedirect.com/science/article/abs/pii/S0304389418301092

Answer: Thank you so much for this point out. The authors think that this suggestion will improve the quality of our work. As the reviewer requested, the importance of bimetallic MOF was discussed and synergic effects between them were discussed.

  1. How the authors predicted the structure of MOF needs to be explained.

Answer: Thank you so much for your contribution. We used the previous method that was used to synthesize MOF materials with different methods that indicate the various shape of materials. We mentioned the synthesis procedure of MOF materials in the electronic supporting information with details.

  1. Why the authors have not performed the XPS analysis to know the oxidation states of bimetals?

Answer: Thank you so much for this suggestion. Because of Covid-19 conditions, it can not be sent our composite to characterize because as we know Turkey has just one XPS device and it didn’t work when we did this study and now it is not accessible.

  1. Why authors selected Co and Ni-based MOF

Answer: Thank you so much for your valuable time and contribution. We chose this composite because of several reasons such as the high conductivity of this composite to increase the sensitivity of the developed electrode and a wide active surface area that it can be modified through MIP polymers to enhance the selectivity of the developed electrode.

  1. Is the oxidation potential of Co and Ni affecting the FAV oxidation reactions needs to be discussed.

Answer: Thank you so much for this point out. As the dear reviewer requested, we discussed the advantage of bimetallic MOFs compared to mono metallic MOFs. The proportion of Co and Ni elements has been tuned to improve the morphology and chemical and electrochemical properties of the material, resulting in enhanced catalytic and analytical performance compared to mono-metal MOFs. However, we can not observe the oxidation effect of Co and Ni separately because we do not have that device and as you know it is related to physics.

  1. SEM results are good but further discussion is necessary related to the bimetallic presence.

Answer: Thank you so much for your comments and valuable time. We discussed more in the EDX that approve the successful synthesis of Co/Ni@MOF materials with this procedure.

  1. Respective Bode plot for Fig.4b need to be shown.

Answer: Thank you so much for your contribution and this point out. We wrote Fig. 4B in the manuscript.

  1. What’s the use to detect FAV in river water samples needs to be explained.

Answer: Thank you so much for this point out. We wrote this sentence to approve the impotance of FAV detection in water samples. FAV undergoes renal excretion, eliminated in the urine mostly as a hydroxide, resulting in the plasma levels of this drug being challenging to control due to its once-daily dosing regimen. Consequently, the ability to accurately detect FAV in the environment and biological samples is increasingly important. However, I revised this sentence and the biological sample was also added in the revised version. As we mentioned in the prior sentence, because of the presence of FAV in human urine after use, this drug will release in the environment such as water; so water sample was chosen as real samples.

  1. What is the active surface area of MOF-based electrodes?

Answer: Thank you so much for your great attention. Since we used CV and DPV to approve and compare the different electrode surfaces (Fig. 4 and Fig. 3), indicating that Co/Ni@MOF nanosheet has more electroactivity compared to other materials. These results showed high current density of Co/Ni@MOF nanosheets. Therefore, we chose this shape and we did not measure the electro active surface area for that.

  1. Why do the authors use CV over DPV?

Answer: Thank you so much for your valuable time. As you know, the sensitivity and selectivity of the DPV method is completely higher than CV. Therefore, we selected DPV for further experiments.

  1. The conclusion needs to be elaborated.

Answer: Thank you so much for your great suggestion. As the dear reviewer requested, the conclusion was revised accordingly.

  1. Cross-check the references

Answer: Thank you so much for your attention. The conclusion was checked again accordingly.

I feel the theme of the article is interesting and attractive.

Round 2

Reviewer 2 Report

The authors made sufficient changes article can be acceptable in the present form. 

Author Response

Thank you so much for your contribution and valuable time